# Breaking the Trilemma of Privacy, Utility, Efficiency via Controllable Machine Unlearning

## ABSTRACT

Machine Unlearning (MU) algorithms have become increasingly critical due to the imperative adherence to data privacy regulations. The primary objective of MU is to erase the influence of specific data samples on a given model without the need to retrain it from scratch. Accordingly, existing methods focus on maximizing user privacy protection. However, there are different degrees of privacy regulations for each real-world web-based application. Exploring the full spectrum of trade-offs between privacy, model utility, and runtime efficiency is critical for practical unlearning scenarios. Furthermore, designing the MU algorithm with simple control of the aforementioned trade-off is desirable but challenging due to the inherent complex interaction. To address the challenges, we present **Con**trollable **M**achine **U**nlearning (ConMU), a novel framework designed to facilitate the calibration of MU. The ConMU framework contains three integral modules: an important data selection module that reconciles the runtime efficiency and model generalization, a progressive Gaussian mechanism module that balances privacy and model generalization, and an unlearning proxy that controls the trade-offs between privacy and runtime efficiency. Comprehensive experiments on various benchmark datasets have demonstrated the robust adaptability of our control mechanism and its superiority over established unlearning methods. ConMU explores the full spectrum of the Privacy-Utility-Efficiency trade-off and allows practitioners to account for different real-world regulations. Source code available at: https://anonymous.4open.science/r/ConMU-B004/

## KEYWORDS

Machine Unlearning, Data Privacy, Trustworthy ML, Deep Learning

**ACM Reference Format:**
Anonymous Author(s). 2023. Breaking the Trilemma of Privacy, Utility, Efficiency via Controllable Machine Unlearning. In *Proceedings of The Web Conference 2024 (WWW '24)*. ACM, New York, NY, USA, 12 pages. https://doi.org/10.1145/nnnnnnn.nnnnnnn

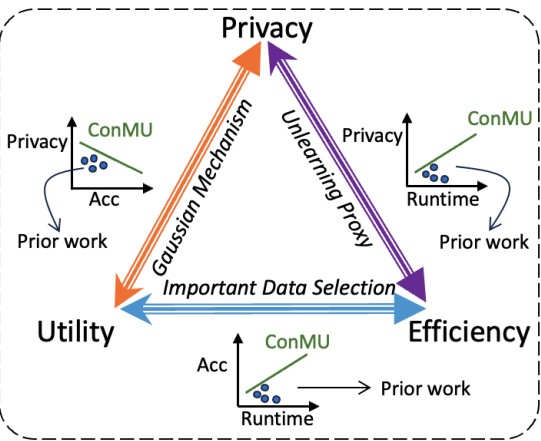

**Figure 1: Privacy, utility, efficiency trilemma in machine unlearning. All previous works have focused on either one or two extremities of the problem while ignoring the full spectrum of trade-offs between the trinity (as shown in blue dots on each subplot). Each of the proposed modules in our ConMU offers smooth control of a pair of two unlearning aspects specifically. Together, ConMU is capable of achieving a satisfactory outcome for versatile practical scenarios, including various degrees of privacy regulations, efficiency constraints, and utility objectives.**

## 1 INTRODUCTION

Machine Learning (ML) models are often trained on real-world datasets in various domains, including computer vision, natural language processing, and recommender systems [6, 10, 32, 58]. For example, many computer vision models are trained on images provided by Flickr users [55], whereas an amount of natural language processing and recommender system algorithms have a high reliance on IMDB [40]. Meanwhile, privacy regulations like the General Data Protection Regulation of the European Union [33] and the California Consumer Privacy Act [46] have established the *right to be forgotten* [5, 17, 44]. This mandates the elimination of specific user data from models upon removal requests.

One naive approach to "forget" user data is removing it from the training set. However, this cannot provide sufficient privacy since ML models tend to memorize training samples [23]. Organizations must either expensively retrain the model from scratch without using specified samples or employ *machine unlearning* [9] techniques to protect user data privacy. Machine unlearning methods are designed to meticulously eliminate samples and their associated influence from both the dataset and the trained model. This safeguards the data privacy with unlearning, protecting it from potential malicious attacks and privacy breaches.

Beyond privacy, utility and efficiency are also important aspects of machine unlearning problems. For instance, sacrificing utility by naively returning constant or purely random output ensures privacy but results in a useless model. On the other hand, retraining from scratch without data subject to removal guarantees privacy and utility yet is prohibitively expensive. Designing a method that

simultaneously maximizes the privacy, utility, and efficiency aspects of machine unlearning is critical and needed. Unfortunately, theoretical machine unlearning research provides evidence that there is an inevitable privacy-utility-efficiency trade-off even for convex problems [13, 30, 45, 48] and similar phenomena exist in other privacy problems [11, 12]. This leads to a trilemma of trinity aspects of machine unlearning, which are accuracy, privacy, and runtime efficiency. While the aforementioned theoretical unlearning solutions provide smooth control among the trinity, they are restricted to simple models and cannot be generalized to general deep-learning approaches.

While a number of efforts have been put into machine unlearning, existing unlearning solutions for deep neural networks mainly focus on maximizing part of the trinity while neglecting their delicate trade-off. In real-world scenarios, different applications would require different levels of privacy regulations, runtime constraints, and utility demand. For example, protecting user identities in healthcare applications [22] might be stricter than safeguarding friendship data on social networks. However, autonomous driving [10] and fraudulent attack detection in financial systems [59] would prioritize accuracy and runtime efficiency more than privacy.

Therefore, a practical machine unlearning solution should be able to easily account for different levels of privacy, utility, and efficiency requirements that arise from various tasks in practice. Unfortunately, the literature lacks a comprehensive examination and controllable MU approach for deep learning to the intricate dynamics involved in balancing privacy, model accuracy, and runtime efficiency. A natural yet pivotal research question arises: "*How to resolve the unlearning trilemma for deep neural networks?*".

To answer this question, we present Controllable Machine Unlearning (ConMU), a novel framework that consists of three components: an important data selection module, a progressive Gaussian mechanism, and an unlearning proxy. Each component emphasizes one part of the aforementioned trilemma, see Figure 1 for the illustration. In particular, the important data selection module modulates the relationship between runtime and model accuracy. The progressive Gaussian mechanism controls the trade-offs between accuracy and privacy. The unlearning proxy facilitates a re-calibration between runtime and privacy. We further underscore that the ConMU is adaptable and can be generalized across diverse model architectures. Among all conducted experiments, ConMU achieves the best privacy performance across 10 out of 12 experiments, with competitive model utility and a 10-15x faster runtime efficiency. Additionally, compared to the naive control baseline, ConMU has illustrated greater control over the trilemma by exhibiting superior and stable performance under the influence of multiple trade-offs. Our main contributions are as follows:

- To the best of our knowledge, this is the first work of tackling the critical trilemma within the realm of machine unlearning for deep neural networks, with a specific focus on the delicate balance between privacy, model utility, and runtime efficiency.
- We propose ConMU, which contains three modules, each designed to reconcile these competing factors: important data selection, progressive Gaussian mechanism, and unlearning proxy.

- Extensive experiments demonstrate the effectiveness of our proposed framework under both class-wise and random forgetting requests.

## 2 RELATED WORK

### 2.1 Machine Unlearning with Theoretical Guarantees

The concept of machine unlearning was first raised in [9]. In general, two unlearning criteria have been considered in previous works: *Exact Unlearning* and *Approximate Unlearning*. Exact unlearning requires eliminating all information relevant to the removed data so that the unlearned model performs exactly the same as a completely retrained model. For example, the authors of [25] presented unlearning approaches for $k$-means clustering. [5] proposed the SISA framework that partitions data into shards and slices, and each shard has a weak learner, which enables quick retraining when dealing with unlearning requests. However, exact unlearning does not allow algorithms to trade privacy for utility and efficiency due to its high requirements for privacy level. In contrast, unlike exact unlearning, approximate unlearning only requires the parameters of the unlearned model to be similar to a retrained model from scratch. Therefore, it is possible for approximate unlearning to sacrifice a portion of the privacy in exchange for better utility and efficiency. [30, 48] studied the approximate unlearning for the cases of linear models and convex losses. [42] extended the idea and provided a theoretical guarantee on weak convex losses. [13, 45] generalize the method of [30] to the graph learning domain. Nevertheless, none of these approximate unlearning solutions apply to general deep neural networks.

### 2.2 Unlearning in Deep Learning Models

Machine unlearning for deep neural networks is challenging because of the non-convex nature of the loss function [3]. [36] approximated the model perturbation towards the empirical risk minimization on the remaining datasets, using the inverse of Hessian. [27] used fisher-based unlearning and introduced an upper bound of SGD-based algorithms to scrub information from intermediate layers of DNNs. [28] extended the framework and introduced forgetting methods using NTK theory [34]. [14] proposed a knowledge adaptation technique where the unlearned model tries to learn from a competent teacher model about retained dataset and an incompetent model about forgetting dataset. [15] proposed an unlearning method without using any training samples, in which they used the error-maximizing noise, proposed by [54], to generate an impaired forgetting dataset, and then used the error-minimizing noise to generate the approximated retained dataset. However, this method yields poor results. Thus, [15] proposed another unlearning algorithm that uses gated knowledge transfer in a teacher-student framework. [56] also proposed a Knowledge Gap Alignment method that minimizes the output distribution difference between models that are trained on different data samples. [53] focuses on deep regression unlearning tasks using a partially trained blindspot model to minimize the distribution difference with the original model. Lastly, [35] showed that applying unlearning algorithms on pruned models gives better performance.

Though important data selections were largely used in deep learning [47], their implementation in the MU field is still unexplored. We discovered that the important data selection is able to offer strong control over the utility-efficiency trade-off. Similarly, Gaussian Noise had been largely adopted in the field of differential privacy [1, 2, 7, 8, 18–20, 24], while its usage in machine unlearning is not yet fully investigated. In addition, the concept of utilizing partially competent teachers for a privacy-efficiency trade-off has not been previously examined. Moreover, many of the existing works focused solely on privacy, overlooking the relationships between accuracy, privacy, and runtime efficiency. Unlike other machine-unlearning algorithms, our method gives users exceptional flexibility and control over the trade-offs among these three factors. In addition, our method imposes no restrictions on optimization methodologies or model architecture.

## 3  PRELIMINARIES

Removing certain training data samples can impact a model's accuracy, potentially improving, maintaining, or diminishing it [52]. As noted by [14], significant discrepancies between unlearned and retrained models can lead to the Streisand effect, inadvertently revealing information about forgotten samples through unusual model behavior. Therefore, the goal of Machine Unlearning is to erase the influence of the set of samples we want to forget so that the unlearned model approximates the retraining one.

Let $D_o = \{x_i\}_{i=1}^N$ be the complete dataset before unlearning requests, in which $x_i$ is the $i^{th}$ sample. Let $D_f$ be the set of samples we want to forget as forgetting dataset, and the complement of $D_f$, which we denote as $D_r$, is the set of samples retained in the training samples, i.e. $D_f \cup D_r = D_o$ and $D_f \cap D_r = \emptyset$. In the setting of random forgetting, $D_f$ may contain samples from different classes of $D_o$. In class-wise forgetting, $D_f$ is a set of examples that have the same class labels. We denote $\theta_o$ as the parameters of the original model, which was trained on $D_o$, denote the parameters of unlearned models as $\theta_u$, and denote the parameters of retrained model $\theta_r$, which is the model completely retrained from scratch using only $D_r$. Lastly, let $\theta_I$ denote the parameters of the unlearning proxy, which has the same model architecture as $\theta_o$, but was only partially trained on $D_r$ for a few epochs.

$\theta_r$ is the gold standard in our MU problem. The goal of machine unlearning is to approximate our $\theta_u$ to $\theta_r$, with less computational overhead. However, for machine unlearning on deep neural networks, achieving a balance between utility, privacy, and efficiency has always been a difficult task.

## 4  METHODS

To address such a trilemma in machine unlearning, we introduce the **ConMU** (Figure 2), a novel framework that consists of an important data selection, a progressive Gaussian mechanism, and an unlearning proxy that modulate relationships among accuracy, privacy, and runtime efficiency. First, the important data selection module (Figure 2 (a)) selectively discards unimportant retaining and forgetting data samples that will not be utilized by subsequent modules. Discarding more samples improves training time while degrading the model's accuracy. Next, the Progressive Gaussian Mechanism module (Figure 2 (b)) injects Gaussian noise into the

remaining forgetting dataset. The amount of noise can control the balance between privacy and accuracy. Subsequently, an unlearning proxy model (Figure 2 (c)) is trained on the retained dataset for a select number of epochs. Through knowledge transfer, the training epoch of the proxy can balance the runtime and privacy. Finally, by fine-tuning the original model using the concatenated retained and noised forgetting datasets, it is transformed into an unlearned version. As a result, by controlling the data volume, the Gaussian noise level, and proxy training duration, we are able to account for different privacy-utility-efficiency requirements. Subsequent sections delve deeper into each module's capabilities and their influence on the trilemma.

### 4.1  Important Data Selection

Unlearning acceleration is crucial in MU. Since our method uses both remaining noised $D_f$ and remaining $D_r$ to perform fine-tuning, the amount of $D_f$ and $D_r$ play significant roles in the run time of our proposed methods. However, the large quantities of $D_f$ and $D_r$ will likely result in an inefficient MU algorithm with a long runtime. Therefore, to facilitate this process, we introduce a novel filtering method using EL2N scores to determine which samples are important for unlearning scenarios. Suppose that $f(\theta, x)$ is the output of the neural network $\theta$ with given data $x$, and denote $y$ as the true class label of $x$. We calculate the mean and the standard deviation of $l2$ normed loss:

$$\mu_\theta(x) = \mathbb{E}_x ||f(\theta, x) - y||_2, \tag{1}$$

$$\sigma_\theta(x) = \sqrt{\mathbb{V}_x ||f(\theta, x) - y||_2}. \tag{2}$$

A higher $\mu_\theta$ means that $x$ is hard to learn and they tend to be the outliers in the dataset. A lower $\mu_\theta$ means that $\theta$ can fit $x$ well. Therefore, we can keep data samples important for the generalization of models by keeping data within a certain range of samples that don't have a very high or low $\mu_\theta$. In our method, we introduce two controllable hyperparameters $z_1$ and $z_2$ and calculate a bound:

$$[\mu_\theta(x) - z_1 \times \sigma_\theta(x), \mu_\theta(x) + z_2 \times \sigma_\theta(x)]. \tag{3}$$

This bound gives users control of how many important data points we want to include by tuning $z_1$ and $z_2$. If we include more data, our accuracy increases, but the runtime also increases. As a result, the ConMU can have a greater speed-up while maximally preserving accuracy by utilizing important data samples.

### 4.2  Progressive Gaussian Mechanism

MU algorithms aim to erase the information about $D_f$ from the original model. In order to forget $D_f$, we can continue training the original model using an obfuscated version of $D_f$, prompting catastrophic forgetting of $D_f$. Within this context, we propose the progressive Gaussian mechanism, which leverages Gaussian noise to obscure the selected $D_f$. Moreover, one of the standout features of this approach is that the magnitude and the shape of Gaussian noise applied to the dataset serve as tunable hyperparameters, granting a remarkable degree of control over the process. More formally, after selecting a subset of important samples:

$$D'_f \in [\mu_{\theta_u}(D_f) - z_1 \times \sigma_{\theta_u}(D_f), \mu_{\theta_u}(D_f) + z_2 \times \sigma_{\theta_u}(D_f)], \tag{4}$$

the ConMU adds Gaussian noise to data samples to balance privacy and accuracy. More specifically, for each data samples in $D'_f$, we

Figure 2: The overall framework of proposed method ConMU, which is placed after forgetting request. In (a), an important data selection is implemented to select data samples that are important to the model. A customized upper/lower bound is attached to this module to facilitate the selection process. Then, the selected forgetting data $D'_f$ is passed to (b), the progressive Gaussian mechanism, to gradually inject Gaussian noise. More noise in the image leads to higher privacy. Afterward, the processed forgetting data $D''_f$ is concatenated with the selected retaining data $D'_r$, which is used for fine-tuning the original model. The unlearning proxy (c) is partially trained on the retaining data $D_r$ and knowledge is transferred to the original model via KL Divergence.

add Gaussian noise and obtain:

$$D''_f = D'_f + \alpha \times N, \ N \sim \mathcal{N}(\mu, \sigma^2 \mathbf{I}), \qquad (5)$$

where $\alpha$, $\mu$, and $\sigma^2$ are controllable hyperparameters, where $\mu$ and $\sigma^2$ represent the mean and variance of the Gaussian distribution, and the $\alpha$ represents the number of times the noise was added to the sample. With more noise being added to the data samples, we will get higher privacy, but lower model accuracy. Therefore, the progressive Gaussian mechanism controls the amount of information they want to scrub away and the amount of information that they want to preserve to maintain the accuracy of the model. In Section 5.3, we empirically demonstrated that with larger $\alpha$, the accuracy decreases and the privacy increases, and vice versa.

### 4.3 Fine-tuning with Unlearning Proxy

The objective of machine unlearning is to align the output distribution of the unlearned model closely with that of the retrained model — a model never exposed to the forgotten data samples. To achieve this, we can utilize an unlearning proxy model, which is a model that has the same architecture as the original model and is partially trained on the retained dataset for a few epochs. By transferring the knowledge of the behavior of the unlearning proxy, we can obtain an unlearned model that contains less information about the forgetting datasets.

More formally, the unlearning proxy model $\theta_I$ is partially trained on the retained dataset $D_r$ for $\delta$ epochs, in which $\delta$ is a hyperparameter. Next, we compute the KL divergence between the probability

distribution of $\theta_I$'s output on the input data $x$ and that of the $\theta_u$ as:

$$D_{KL}(\theta_I(x) \parallel \theta_u(x)) = \sum_i \theta_I(x)(i) \log \left( \frac{\theta_I(x)(i)}{\theta_u(x)(i)} \right), \qquad (6)$$

where $i$ corresponds to the data class. We want to minimize this KL divergence, aiming to make the output distribution of the unlearned model $\theta_u$ as close as possible to that of a model that has never seen $D_f$, which is the unlearning proxy. In section 5.3, we demonstrate that if $\delta$ increases, the $\theta_u$ will become more similar to $\theta_r$, but with increasing runtime.

### 4.4 Controlling Machine Unlearning

After discussing the individual modules for important data selection, progressive Gaussian mechanism, and using an unlearning proxy, we now focus on how these parts come together. First, we obtain $D_{new} = D''_f \cup D'_r$, in which:

$$D'_r \in [\mu_{\theta_u}(D_r) - z'_1 \times \sigma_{\theta_u}(D_r), \mu_{\theta_u}(D_r) + z'_2 \times \sigma_{\theta_u}(D_r)], \qquad (7)$$

and $z'_1$ and $z'_2$ are two hyperparameters for filtering the retained dataset, as discussed in 4.1. With $D_{new}$, we will use the cross-entropy (CE) loss to further train $\theta_u$ on $D_{new}$, combined with the KL loss in section 4.3. The loss to train the unlearned model $\theta_u$ is defined as:

$$\mathcal{L} = CE(D_{new}) + \gamma D_{KL}(\theta_I(D_{new}) \parallel \theta_u(D_{new})). \qquad (8)$$

The $\gamma$ in Equation (8) ensures that these two losses are on the same scale. In summary, the ConMU uses Equation (8) to fine-tune the

original model $\theta_o$ to achieve $\theta_u$, with way fewer epochs required by complete retraining, allowing the calibration of the amount of data to preserve, the amount of noise added to the filtered forget data samples, and number of times to train the unlearning proxy. With these three modules, the ConMU allows controllable trade-offs between accuracy, privacy, and runtime.

## 4.5 Forget-Retain-MIA Score

There are many evaluation metrics to determine the privacy of the unlearning algorithms. For example, many literatures used Retain Accuracy (RA) and Forget Accuracy (FA) [4, 14, 15, 26, 27, 35, 44, 54], which are the generalization ability of the unlearned model on $D_r$ and $D_f$, respectively. Moreover, many previous works have used Membership Inference Attacks (MIA) [14, 21, 29, 35, 44, 51] that determine whether a particular training sample was present in the training data for a model.

Given this landscape of varied metrics, it becomes imperative to consolidate them to yield a more comprehensive evaluation. As we have stated in section 3, our goal for the evaluation of privacy is to ensure minimal disparity between our model's outcomes and the retrained model, which is the gold standard of unlearning tasks. Therefore, we introduce a new evaluation metric called the *Forget-Retain-MIA* (**FRM**) score that considers the differences between the unlearned model with the retrained model on the trifecta of FA, RA, and MIA, which is inspired by NeurIPS 2023 machine unlearning challenge [1]. Suppose we denote $FA_r$, $RA_r$, and $MIA_r$ as the FA, RA, and MIA performance of the retrained model, and denote $FA_u$, $RA_u$, and $MIA_u$ as the FA, RA, and MIA performance of the unlearning model, we calculate the FRM score as:

$$FRM = exp\big(-\big(\frac{|FA_u - FA_r|}{FA_r} + \frac{|RA_u - RA_r|}{RA_r} + \frac{|MIA_u - MIA_r|}{MIA_r}\big)\big).$$
(9)

The FRM score quantitatively compares the normalized differences in FA, RA, and MIA performances of the unlearning model with that of its retrained counterpart. The FRM score lies between 0 and 1. An FRM score of an unlearning model will be closer to 1 if the unlearned model's privacy is perfectly aligned with the retained model's privacy, and it will be closer to 0 if the model is completely different from the retrained model. An ideal FRM score of 1 signifies that the unlearning algorithm has achieved exact unlearning. We use the FRM score to evaluate the ConMU and other baseline models' performance on privacy in the subsequent experiment sections.

## 5 EXPERIMENTS

In this section, we conduct extensive experiments to validate the effectiveness of the ConMU. In particular, through the experiments, we aim to answer the following research questions: (1) Can ConMU find the best balance point given the trilemma? (2) Can each module effectively control a specific aspect of the trilemma? (3) Can the naive fine-tune method possess the same control ability as the ConMU?

[1]https://unlearning-challenge.github.io/

## 5.1 Experiment setups

*5.1.1 Datasets and models.* Our experiments mainly focus on image classification for CIFAR-10 [37] on ResNet-18 [31] under two unlearning scenarios: random data forgetting and class-wise data forgetting. Besides, additional experiments are conducted on CIFAR100 [37], and SVHN [43] datasets using vgg-16 [50]. The details of the dataset are shown in Appendix C.

*5.1.2 Baseline Models.* For baselines, we compare with Fine-Tuning (FT) [27, 35, 57], Gradient Ascent (GA) [29, 35], and Influence Unlearning (IU) [16, 26, 35, 36, 44]. In particular, FT directly utilizes retained dataset $D_r$ to fine-tune the original model $\theta_o$. The GA method attempts to add the gradient updates on $D_f$ during the training process back to the $\theta_o$. Lastly, IU leverages influence functions to remove the influence of the target data sample in $\theta_o$. Besides, [35] has shown that pruning first before applying unlearning algorithms will increase performance. Therefore, we apply the OMP (one-shot-magnitude pruning) [35, 38, 39, 41] to each baseline model as well as the ConMU. The details of each baseline model are elaborated in Appendix B.1.

*5.1.3 Evaluation Metrics.* We aim to evaluate MU methods from five perspectives: test accuracy *forget accuracy (TA)*, *forget accuracy (FA)*, *retain accuracy (RA)*, *membership inference attack (MIA)* [49], *runtime efficiency (RTE)*, and *FRM privacy score*. Specifically, TA measures the accuracy of $\theta_u$ on the testing datasets and evaluates the generalization ability of MU methods. FA and RA measure the accuracy of the unlearned model on forgetting dataset $D_f$ and retaining dataset $D_r$, respectively. MIA verifies if a particular training sample existed in the training data for the original model. Lastly, we use the *FRM privacy score* metric to comprehensively evaluate the privacy level of an MU method. Additional details of evaluation metrics are illustrated in Appendix A.

*5.1.4 Implementation Details.* We report the mean of ten independent runs with different data splits and random seeds. For random forgetting, we randomly selected 20% of the training samples as forgetting datasets. For class-wise forgetting, we randomly selected 50% of a particular class for different datasets as the forgetting samples. The details will be in appendix B.2.

## 5.2 Experiment Results

To answer the first question: **Can ConMU find the best trade-off points given three important factors?** We conduct random forgetting and class-wise forgetting to comprehensively evaluate the effectiveness of a MU method. The performance is reported in Table 1. Note that a better performance of a MU method contains a smaller performance gap with the retrained model (except RTE and TA), which is the gold standard for MU tasks. According to the table, we can find that IU (influence unlearning) and GA (gradient ascent) with OMP pruning achieve satisfactory results under unlearning privacy (FA, RA, MIA, and FRM) and efficiency (RTE) metrics with relatively shorter runtime. According to the table, IU is usually the fastest baseline, while GA is the runner-up. However, this outstanding unlearning efficiency comes at a high cost to the model utility, rendering them the worst baseline models in terms of test accuracy. Alternatively, FT (fine-tuning) performs well across all metrics with the exception of unlearning efficiency.

**Table 1: Overall results of our proposed ConMU, with a number of baselines under two unlearning scenarios: random forgetting and class-wise forgetting. Since a retrained model is golden for unlearning tasks, we evaluate the performance of MU models based on their similarities to the retrained model. Bold indicates the best performance and underline indicates the runner-up. The unlearning performance of each MU method is evaluated under five metrics: test accuracy (*TA*), accuracy on forget data (*FA*), accuracy on retain data (*RA*), membership inference attack (*MIA*), and running time efficiency (*RTE*). An additional *FRM* is added on top of that to thoroughly evaluate the privacy level of each method. Note that a larger value of FRM denotes a closer privacy level as a retrained model. A performance difference against the retrained model is reported in (●). A better performance in the metrics with blue ↓ has the smallest gap with retrained model. While the metrics with black ↑ favor a greater performance value.**

| MU Methods | TA(%) ↑ | FA(%)↓ | RA(%)↓ | MIA(%)↓ | RTE(s) | FRM Privacy ↑ |
|---|---|---|---|---|---|---|
| | | Resnet-18 Random data forgetting (CIFAR-10) | | | | |
| retrain | 79.99 | 80.46 (0.00) | 91.47 (0.00) | 0.196 (0.00) | 933.51 | 1 |
| IU + Pruning | 41.63 | 41.62 (38.84) | 41.21 (50.26) | 0.576 (0.38) | 33.69 | 0.051 |
| GA + Pruning | 64.61 | 66.74 (13.72) | 66.15 (25.32) | 0.341 (0.145) | **28.15** | 0.305 |
| FT + Pruning | **84.71** | 84.13 (3.67) | **90.96 (0.51)** | 0.154 (0.042) | 475.99 | 0.767 |
| ConMU | 78.83 | **81.22 (0.76)** | 81.75 (9.72) | **0.189(0.008)** | 59.59 | **0.855** |
| | | Resnet-18 Class-Wise forgetting (CIFAR-10) | | | | |
| retrain | 82.55 | 68.22 (0.00) | 89.67 (0.00) | 0.339 (0.00) | 1241.92 | 1 |
| IU + Pruning | 20.39 | 0.01 (68.21) | 20.96 (68.71) | 1.00 (0.661) | 40.15 | 0.024 |
| GA + Pruning | 52.22 | 15.22 (53) | 53.88 (35.79) | 0.832 (0.493) | **25.24** | 0.072 |
| FT + Pruning | **85.75** | 69.89 (1.67) | **92.10 (2.43)** | 0.278 (0.061) | 565.47 | 0.793 |
| ConMU | 83.61 | **67.23 (0.99)** | 86.68 (5.75) | **0.329 (0.01)** | 89.4 | **0.925** |
| | | VGG Random data forgetting (CIFAR-10) | | | | |
| retrain | 81.10 | 81.49 (0.00) | 92.09 (0.00) | 0.195 (0.00) | 881.57 | 1 |
| IU + Pruning | 59.74 | 57.97 (23.52) | 57.52 (34.57) | 0.393(0.198) | 38.36 | 0.186 |
| GA + Pruning | 69.43 | 69.97(11.52) | 69.79 (22.30) | 0.292 (0.097) | 47.17 | 0.414 |
| FT + Pruning | **83.88** | 58.98(22.51) | **90.64 (1.45)** | 0.384 (0.189) | 378.02 | 0.283 |
| ConMU | 79.09 | **82.52 (1.03)** | 84.00 (8.09) | **0.175(0.02)** | **32.42** | **0.816** |
| | | VGG Class-Wise forgetting (CIFAR-10) | | | | |
| retrain | 82.41 | 69.02 (0.00) | 92.90 (0.00) | 0.334 (0.00) | 1034.40 | 1 |
| IU + Pruning | 53.06 | 27.16 (41.86) | 53.08 (38.82) | 0.655 (0.321) | 46.70 | 0.136 |
| GA + Pruning | 53.18 | 11.96 (57.06) | 54.51 (38.39) | 0.864 (0.530) | **30.74** | 0.059 |
| FT + Pruning | **83.88** | 58.98 (10.04) | **90.64 (2.26)** | 0.383 (0.049) | 353.97 | 0.729 |
| ConMU | 81.12 | **63.75 (5.27)** | 87.10 (5.80) | **0.362 (0.028)** | 148 | **0.800** |
| | | VGG Random data forgetting (CIFAR-100) | | | | |
| retrain | 60.65 | 60.54 (0.00) | 92.49 (0.00) | 0.406 (0.00) | 823.15 | 1 |
| IU + Pruning | 7.15 | 5.97 (54.57) | 5.83 (86.66) | 0.066 (0.340) | **38.74** | 0.069 |
| GA + Pruning | 14.71 | 13.96 (46.58) | 14.24 (78.25) | 0.855 (0.449) | 47.42 | 0.066 |
| FT + Pruning | 49.78 | 47.67 (12.87) | 60.70 (31.79) | 0.509 (0.103) | 215.22 | 0.444 |
| ConMU | **55.22** | **61.53 (0.99)** | **71.77 (20.72)** | **0.414(0.008)** | 65.01 | **0.771** |
| | | VGG Class-Wise forgetting (CIFAR-100) | | | | |
| retrain | 63.71 | 65.78 (0.00) | 93.00 (0.00) | 0.382 (0.00) | 1036.40 | 1 |
| IU + Pruning | 7.14 | 3.56 (62.22) | 5.89 (87.11) | 0.044 (0.338) | 37.95 | 0.063 |
| GA + Pruning | 10.68 | 0.44 (65.34) | 10.21 (82.79) | 0.010 (0.372) | **23.01** | 0.057 |
| FT + Pruning | 57.33 | **64.00 (1.78)** | 74.60 (18.40) | 0.364 (0.018) | 403.07 | 0.762 |
| ConMU | **58.78** | 58.39 (7.39) | **80.03 (12.97)** | **0.385 (0.003)** | 53.07 | **0.771** |

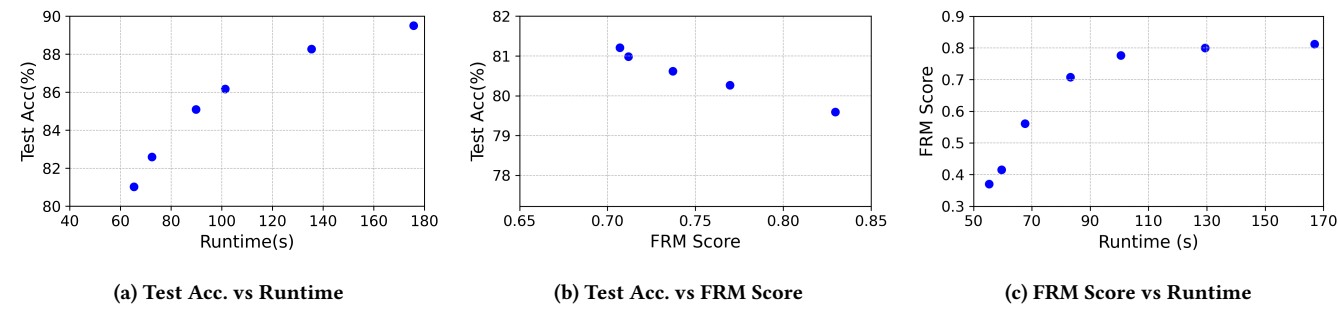

(a) Test Acc. vs Runtime

(b) Test Acc. vs FRM Score

(c) FRM Score vs Runtime

**Figure 3: Ablation study results of each module on CIFAR-10 with ResNet-18. For every module, we fix the other two novel modules while adjusting its own controllable parameters. Since each proposed module is designed to control one side of the trilemma, we present the results for each module in a chart with $x$ and $y$ axes representing their respective controlled factors.**

As shown in Table 1, FT is the runner-up in the majority of cases and achieves a high FRM across all benchmarks. However, this comes with a high sacrifice on runtime efficiency, making it the slowest baseline method. Finally, we observe that the ConMU can outperform other baselines by remarkable margins and achieve a good balance on all privacy metrics and competitive accuracy across CIFAR-10, CIFAR-100, and SVHN, respectively. Additionally, the ConMU has the highest FRM score among all baseline models with an acceptable runtime efficiency relative to other baselines.

### 5.3 Unlearning Trilemma Analysis

Given our proposed ConMU is to narrow the performance gap with the gold-retrained model and to better control the trade-off between different metrics, we conduct further experiments to validate the effectiveness of each module. The central question addressed is: **Can each module effectively govern a specific facet of the aforementioned trilemma?** The associated results are shown in Figure 3. To better answer this question, we first represent the unlearning trilemma as a triangle (figure 1), wherein each side corresponds to a distinct aspect of the trilemma. An effective control module should identify a balance point anywhere along the side, rather than being confined to the two endpoints. Since the ConMU contains three modules, to better observe the compatibility and flexibility of each module in influencing different metrics, we systematically adjust the input values of each module with random forgetting requests on the CIFAR-10 dataset, showcasing the ability to control trade-offs at various levels.

*5.3.1 Utility vs Efficiency.* In our proposed method, the important data selection module is specifically designed to curate the samples that are later utilized in the fine-tuning process of the pruned model. The rationale behind this is twofold: (1) extracting the samples that contribute significantly to model generalization process, and (2) expediting the runtime of the unlearning process. To further investigate the trade-off between model utility and runtime efficiency, we carefully adjust the upper and lower bounds of the important data selection to incorporate different percentiles of data. In Figure 3 (a), we present the control ability of the proposed important data selection module by selecting different portions of data. We start with the inclusion of 5 % of the data and gradually progress to 90 %. From figure 3 (a), we first discover that a higher percentile of selected data not only prolongs the runtime

but also enhances the utility performance of the ConMU. For instance, increasing the data percentile from 5 % to 25% results in a 7.89 % increase in model accuracy from 75.09 to 81.02. However, this improvement comes at the cost of a 68.17 % increase in runtime, escalating from 38.86 seconds to 65.35 seconds. Furthermore, we observe diminishing returns as the included data percentage increases. Take the last two points as an example, including 10 % more data leads to a mere 1.39 % increase in model utility but incurs a substantial 29.7 % increase in runtime, escalating from 135.43 seconds to 175.71 seconds. This phenomenon suggests that beyond a certain threshold of included data, sacrificing runtime yields only marginal improvements in model utility.

*5.3.2 Utility vs Privacy.* In the intricate landscape of the trilemma, another crucial facet involves the delicate equilibrium between utility and privacy. As mentioned in Section 4.2, the purpose of such a module is to disrupt the forgetting information in samples, where a higher noise level indicates that the sample contains more chaotic information, which represents better privacy. To validate this hypothesis, we modify the mechanism's noise level to demonstrate the relationship between model utility and privacy. Figure 3 (b), illustrates the performance of the proposed progressive Gaussian mechanism module under varying noise levels. We begin with a noise level of 0, which uses the selected data from the previous module, and increase it to a noise level of 10. As shown by the increasing FRM score, a higher noise level results in a closer privacy level with respect to the retrained model (higher FRM score). For example, increasing the noise level from 0 to 2 increases the FRM by 4.2 %, from 0.707 to 0.737. This enhancement, however, comes with a 0.72 % decrease in model utility, from 81.21% to 80.62%. This trend is consistent as the intensity of noise increases. As the noise level increases from 8 to 10, the model test accuracy decreases from 79.59 % to 78.22%, a decrease of 1.75 %, while the FRM increases by 3.2 %. This phenomenon demonstrates the viability of the compromise between model utility and privacy.

*5.3.3 Privacy vs Efficiency.* Lastly, there is a discernible trade-off between model privacy and runtime performance. As mentioned in Section 4.3, we introduce an unlearning proxy to strike a balance between these two crucial factors. This module's purpose is to reduce the privacy disparity between the retrained model and

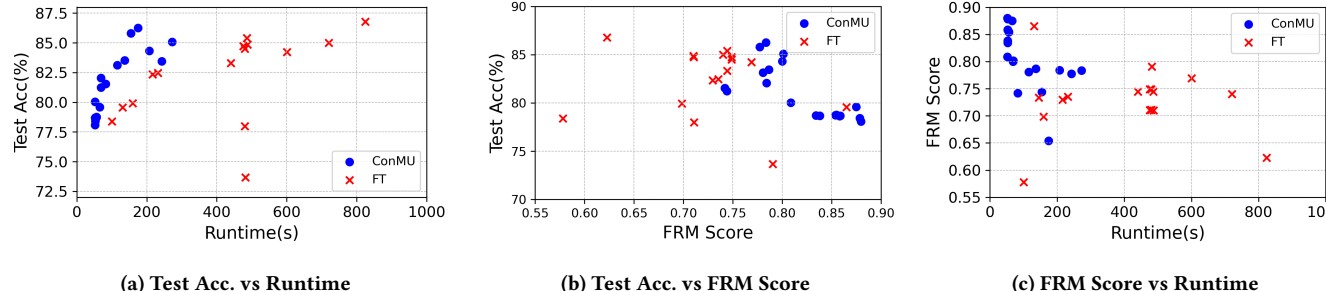

(a) Test Acc. vs Runtime        (b) Test Acc. vs FRM Score        (c) FRM Score vs Runtime

Figure 4: Unlearning performance of ConMU and Naive Fine-tuning on CIFAR-10 with ResNet-18 under random forgetting request using various combinations of controllable mechanisms. For FT, we adjust the epoch number from 5 to 35 and try different learning rates ranging from 0.1 to 0.0001. Figure 4 (a) focuses on evaluating the relationship between utility and runtime efficiency, where $x$ and $y$ axes denote the test runtime and accuracy, respectively. Figure 4 (b) focuses on the relationship between utility and privacy, where $x$ and $y$ axes denote the FRM score and test accuracy, respectively. Figure 4 (c) depicts the relationship between privacy and runtime efficiency, where $x$ and $y$ axes denote the runtime and FRM score, respectively. The red point represents the performance of the fine-tuning method and the blue point denotes the ConMU.

the unlearning model by means of an unlearning proxy. To validate this effect, we progressively increase the training epoch of the unlearning proxy from 0 to 8, enabling it closer to the retrained model. As shown in Figure 3 (c), an increase in the number of training epochs in the unlearning proxy resulted in improved privacy performance, as indicated by a higher FRM score. Raising the proxy training epoch from 2 to 3 increases runtime by 23.13 %, from 67.61 seconds to 83.25 seconds. This results in a 26.02 % increase in FRM score. However, this progress diminishes when FRM exceeds 0.75. Consider the last three data points as an illustration: a 66.1 % increase in duration from 100.47 seconds to 166.89 seconds results in a 4.64 % increase from 0.776 to 0.812. Similarly to the trade-off between utility and runtime, there exists a threshold between privacy and runtime where sacrificing one does not result in a substantial improvement in the other.

## 5.4 ConMU vs. Naive Fine-Tune Method

The overall performance of the rudimentary fine-tune (FT) baseline method, as shown in Table 1, is comparable to that of the ConMU baseline method. Consequently, an intriguing question may be posed: **Can the naive FT model have the same control ability over these trilemmas as the method demonstrated by merely adjusting its hyperparameters?** In order to answer this question, we compare the control ability of the ConMU and naive fine-tuning. To demonstrate this distinction in a holistic manner, we evaluate the performance of two models based on three crucial factors: privacy (FRM), utility (TA), and efficiency (runtime). We primarily demonstrate the control ability of the naive fine-tuning method by varying two parameters: learning rate and fine-tuning epochs. For the ConMU, we alter those three proposed modules.

Figure 4 (a) demonstrates the trade-off between the runtime of each sample and the utility. In addition, figure 4 (b) illustrates the trade-off between utility and privacy, in which greater $x$ values indicate a higher FRM score, which corresponds to a closer level of privacy with the retrained model. As demonstrated in Section 5.3, an expected trade-off would emerge as $x$ values increase from left to right. Figure 4 (c) demonstrates the relationship between privacy and runtime efficiency. Ideally, a sample that resolves the

trilemma should be placed in the top left corner of (a), the top right corner of (b), and the top left corner of (c). Given a similar level of test accuracy, ConMU can achieve a higher FRM score with a shorter runtime. When the test accuracy for FT is 77.99 % and the ConMU is 78.08 %, for example, the FRM is 0.71 and 0.87, respectively. Meanwhile, the runtime is 480.14 seconds and 52.12 seconds, respectively, which is 9x faster. Furthermore, as test accuracy improves, the performance of the ConMU remains relatively stable and consistent. For instance, when the test accuracy for FT increases from 84.22 % to 85 %, the FT's FRM falls from 0.77 to 0.74. In contrast, the FRM for ConMU increases by 0.3 % from 0.799 to 0.801 when the test accuracy increases from 84.32 % to 85.09 %. In terms of the runtime, the FT increases from 600.22 seconds to 720.19 seconds, whereas the ConMU only increases by only 65.55 seconds, from 207.13 to 272.68 seconds. Throughout the resulting chart, ConMU displays its superiority not only in the stability of controlling the trilemma but also a significant margin over the overall performance.

## 6 CONCLUSION

In this paper, we identify the trilemma between model privacy, utility, and efficiency that exists in machine unlearning for deep neural networks. To address this issue and gain greater control over this trilemma, we present ConMU, a novel MU calibration framework. Specifically, ConMU introduces three control modules: the important data selection, the progressive Gaussian mechanism, and the unlearning proxy, each of which seeks to calibrate portions of the MU trilemma. Extensive experiments and in-depth studies demonstrate the superiority of the ConMU across multiple benchmark datasets and a variety of unlearning metrics. Future work could focus on extending our control mechanism to other fields of study, such as the NLP and Graph domain.

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

## Appendix A    EVALUATION METRICS

**Retain Accuracy (RA) and Forget Accuracy (FA)** [4, 14, 15, 26, 27, 35, 44, 54]: Retain Accuracy is the generalization ability of $\theta_u$ on $D_r$, which is referred as the fidelity of machine unlearning. The Forget Accuracy, on the other hand, is the generalization ability of $\theta_u$ on $D_f$, which is the efficacy of machine unlearning. A better $D_f$ score for an approximate unlearning method should minimize the disparity with the retrained model, which is the gold-standard.

**Test Accuracy (TA)**: Test Accuracy (TA), unlike $D_f$ or $D_r$, assesses the generalization ability of $\theta_u$ on the test dataset after unlearning. Besides the task of class-wise forgetting, in which the forgetting class is excluded from the testing dataset, the accuracy of the test is determined using the entire testing dataset.

**Confidence-based Membership Inference Attack (MIA)** [14, 21, 29, 35, 44, 51]: Membership inference attacks determine whether a particular training sample was present in the training data for a model. In our work, we employed confidence-based MIA based on [35]. Formally, we first train an MIA predictor using $D_r$ and test sets. Then, we apply the trained MIA predictor to $\theta_u$ on $D_f$, and we get the following score called MIA-efficacy: $TN/|D_f|$. Here, $TN$ is the number of forgetting samples that are predicted as non-training samples. A higher MIA-efficacy score indicates that $\theta_u$ has less information about $D_f$.

**Runtime Efficiency (RTE)** [9, 35, 44]: Runtime efficiency measures the computational efficiency of an MU method. In Machine Unlearning, we want to achieve the $\theta_r$-like model in less computational time.

## Appendix B    IMPLEMENTATION DETAILS

### B.1    Baseline Descriptions

**Influence unlearning** [16, 26, 35, 36, 44]: The influence function is a technique that allows us to understand how model parameters change by up-weighting training data points, and the effect of these data points can be estimated in closed form. In machine unlearning, we can estimate $|\theta_o - \theta_r|$ if $D_r$ is removed from $D_o$ using influence functions. Also, influence unlearning assumes that the empirical risk is twice-differentiable and strictly convex in $\theta_o$. However, for deep neural networks, due to the non-convexity of the loss function and approximating the inverse-Hessian vector product can be erroneous [3, 36, 44], using influence functions to approximate $\theta_r$ is not effective in practice.

**Gradient Ascent** [29, 35]: Gradient Ascent is doing exactly the opposite of what gradient descent tries to do. It reverses the model training on $D_f$ during training back to the $\theta_o$. To be more specific, gradient ascent approaches move $\theta_o$ towards loss for data points to be erased.

**Regular Fine-tuning** [27, 35, 57]: Different from naively retraining from the scratch, FT fine-tunes the the pre-trained $\theta_o$ on $D_r$ for a few epochs, with a slightly larger learning rate. The motivation is that fine-tuning on $D_r$ may cause catastrophic unlearning over $D_f$.

### B.2    Hyperparameters

The hyper-parameters of ConMU are listed in Table 2 for random forgetting and Table 3 for class-wise forgetting. For random forgetting, we randomly selected 20% of the original training data as the forgetting dataset. For class-wise forgetting, we randomly selected 50% of a particular class as the forgetting data. Due to the limitation of the GPU memory, the batch size is restricted to 128. Note that the $\alpha$ is generally larger for class-wise forgetting than that for random forgetting.

**Table 2: Hyper-parameters of ConMU for *CIFAR-10, CIFAR-100* and *svhn* datasets on ResNet18 and VGG for random forgetting. The $z_1$ and $z_2$ indicate the lower and upper bounds of important data selection, respectively. The $\alpha$ is the amount of Gaussian noise used in the Progressive Gaussian Mechanism. The Gaussian noise has a mean $\mu$ and a standard deviation $\sigma$. $\delta$ is the number of epochs that the incompetent model was trained on. $\gamma$ is the coefficient of the KL loss term.**

| Model | ConMU (ResNet-18) | | | ConMU (VGG) | | |
|---|---|---|---|---|---|---|
| Dataset | CIFAR-10 | CIFAR-100 | svhn | CIFAR-10 | CIFAR-100 | svhn |
| Fine-Tune Epoch | 5 | 5 | 5 | 5 | 5 | 5 |
| Learning Rate | 1e-2 | 1e-2 | 1e-2 | 1e-2 | 1e-2 | 1e-2 |
| Batch Size | 128 | 128 | 128 | 128 | 128 | 128 |
| Optimizer | SGD | SGD | SGD | SGD | SGD | SGD |
| Retain, $z_2$ | 0.3 | 0.25 | 0.28 | 0.2 | 0.45 | 0.3 |
| Retain, $z_1$ | 0.17 | 0.16 | 0.1 | 0.16 | 0.35 | 0.2 |
| Forget, $z_2$ | 1.0 | 0.2 | 0.4 | 0.2 | 0.4 | 0.4 |
| Forget, $z_1$ | 0.85 | 0.18 | 0.3 | 0.15 | 0.37 | 0.3 |
| $\alpha$ | 3 | 4 | 5 | 3 | 5 | 5 |
| $\mu$ | 0 | 0 | 0 | 0 | 0 | 0 |
| $\sigma$ | 1 | 1 | 1 | 1 | 1 | 1 |
| $\delta$ | 1 | 1 | 1 | 1 | 1 | 1 |
| $\gamma$ | 0.5 | 0.5 | 0.5 | 0.5 | 0.5 | 0.5 |

**Table 3: Hyper-parameters of ConMU for *CIFAR-10, CIFAR-100* and *svhn* datasets on ResNet18 and VGG for class-wise forgetting. The forgotten Class is the class index of the dataset we chose to forget for the experiments. The $z_1$ and $z_2$ indicate the lower and upper bounds of important data selection, respectively. The $\alpha$ is the amount of Gaussian noise used in the Progressive Gaussian Mechanism. The Gaussian noise has a mean $\mu$ and a standard deviation $\sigma$. $\delta$ is the number of epochs that the incompetent model was trained on. $\gamma$ is the coefficient of the KL loss term.**

| Model | ConMU (ResNet-18) | | | ConMU (VGG) | | |
|---|---|---|---|---|---|---|
| Dataset | CIFAR-10 | CIFAR-100 | svhn | CIFAR-10 | CIFAR-100 | svhn |
| Forgotten Class | 5 | 69 | 5 | 5 | 69 | 5 |
| Fine-Tune Epoch | 5 | 5 | 5 | 5 | 5 | 5 |
| Learning Rate | 1e-2 | 1e-2 | 1e-2 | 1e-2 | 1e-2 | 1e-2 |
| Batch Size | 128 | 128 | 128 | 128 | 128 | 128 |
| Optimizer | SGD | SGD | SGD | SGD | SGD | SGD |
| Retain, $z_2$ | 0.3 | 0.3 | 0.3 | 0.4 | 0.45 | 0.3 |
| Retain, $z_1$ | 0.2 | 0.1 | 0.2 | 0.3 | 0.35 | 0.2 |
| Forget, $z_2$ | 1.0 | 0.4 | 0.8 | 0.8 | 1.0 | 0.8 |
| Forget, $z_1$ | 0.8 | 0.3 | 0.5 | 0.5 | 0.85 | 0.5 |
| $\alpha$ | 12 | 1 | 4 | 20 | 10 | 4 |
| $\mu$ | 0 | 0 | 0 | 0 | 0 | 0 |
| $\sigma$ | 1 | 1 | 1 | 1 | 1 | 1 |
| $\delta$ | 1 | 1 | 1 | 1 | 1 | 1 |
| $\gamma$ | 0.5 | 0.5 | 0.5 | 0.5 | 0.5 | 0.5 |

**Table 4: Additional results of our proposed ConMU, with a number of baselines under two unlearning scenarios: random forgetting and class-wise forgetting. Bold indicates the best performance and underline indicates the runner-up. The efficacy of each MU method is evaluated under five metrics: test accuracy ($TA$), accuracy on forget data ($FA$), accuracy on retain data ($RA$), membership inference attack ($MIA$), and running time efficiency ($RTE$). The performance of ConMU is reported in the form of $a_{\pm b}$, where $a$ is the mean value of independent 10 trial runs and $b$ denotes the standard deviation. A performance difference against the retrained model is reported in (●).**

| MU Methods | $TA(\%)\uparrow$ | $FA(\%)\downarrow$ | $RA(\%)\downarrow$ | $MIA(\%)\downarrow$ | $RTE(s)$ | FRM Privacy $\uparrow$ |
|---|---|---|---|---|---|---|
| | | | Resnet-18 Random data forgetting (CIFAR-100) | | | |
| retrain | 51.45 | 40.82 (0.00) | 99.97 (0.00) | 0.493 (0.00) | 1066.69 | 1 |
| IU + Pruning | 6.48 | 6.03 (38.84) | 6.02 (93.95) | 0.929 (0.436) | 38.36 | 0.069 |
| GA + Pruning | 31.33 | **31.26** (9.56) | 31.64 (68.33) | 0.675 (0.182) | 54.58 | 0.276 |
| FT + Pruning | **56.36** | 54.87 (14.05) | **70.55** (29.42) | 0.451 (0.042) | 282.09 | **0.484** |
| ConMU | 48.97 | 54.95 (14.13) | 61.25 (38.72) | **0.453** (0.04) | **27.83** | 0.443 |
| | | | Resnet-18 Class-Wise forgetting (CIFAR-100) | | | |
| retrain | 56.43 | 60.89 (0.00) | 99.98 (0.00) | 0.391 (0.00) | 1323.98 | 1 |
| IU + Pruning | 6.52 | 8.89 (52.00) | 6.05 (93.93) | 0.94 (0.549) | 39.14 | 0.041 |
| GA + Pruning | 20.18 | 7.78 (53.11) | 19.53 (80.45) | 0.933 (0.542) | **22.57** | 0.047 |
| FT + Pruning | **60.07** | **76.67** (15.78) | 73.87 (26.11) | 0.244 (0.147) | 364.23 | 0.408 |
| ConMU | 55.22 | 61.53 (28.27) | **71.77** (0.023) | 0.414 (0.023) | 48.23 | **0.704** |
| | | | VGG Random data forgetting (SVHN) | | | |
| retrain | 93.53 | 93.47 (0.00) | 99.62 (0.00) | 0.065 (0.00) | 840.33 | 1 |
| IU + Pruning | 90.18 | 92.30 (1.17) | 91.96 (7.66) | 0.08 (0.015) | 49.30 | **0.726** |
| GA + Pruning | 92.26 | **94.34** (0.87) | 94.38 (5.24) | 0.94 (0.875) | 26.40 | 0 |
| FT + Pruning | **94.19** | 95.71 (2.24) | **99.65** (0.03) | 0.043 (0.022) | 280.15 | 0.696 |
| ConMU | 90.95 | 91.93 (1.54) | 92.51 (7.11) | **0.081** (0.016) | 79.71 | 0.716 |
| | | | VGG Class-Wise forgetting (SVHN) | | | |
| retrain | 94.12 | 90.21 (0.00) | 99.55 (0.00) | 0.098 (0.00) | 1048.65 | 1 |
| IU + Pruning | 90.08 | 92.83 (2.62) | 91.92 (7.63) | 0.072 (0.026) | 35.01 | 0.624 |
| GA + Pruning | 81.38 | 31.64 (58.57) | 86.45 (13.10) | 0.684 (0.586) | 27.99 | 0.057 |
| FT + Pruning | **95.02** | 94.87 (4.66) | **99.99** (0.44) | 0.051 (0.047) | 320.77 | 0.3772 |
| ConMU | 92.31 | **88.43** (1.78) | 94.73 (4.82) | **0.106** (0.008) | 85.05 | **0.864** |
| | | | Resnet-18 Random data forgetting (SVHN) | | | |
| retrain | 91.05 | 91.68 (0.00) | 99.45 (0.00) | 0.083 (0.00) | 954.66 | 1 |
| IU + Pruning | 35.04 | 36.45 (55.23) | 36.78 (62.67) | 0.637 (0.554) | **40.52** | 0 |
| GA + Pruning | 85.35 | 87.87 (3.81) | 87.51 (11.94) | 0.121 (0.038) | 57.55 | 0.538 |
| FT + Pruning | 92.95 | 93.34 (1.66) | **100.00** (0.55) | 0.066 (0.017) | 729.64 | 0.795 |
| ConMU | **92.98** | **93.05** (1.37) | 93.84 (5.61) | **0.07** (0.013) | 74.36 | **0.796** |
| | | | Resnet-18 Class-Wise forgetting (SVHN) | | | |
| retrain | 91.40 | 83.73 (0.00) | 98.71 (0.00) | 0.162 (0.00) | 1075.59 | 1 |
| IU + Pruning | 32.68 | 13.14 (70.59) | 35.48 (63.23) | 0.868 (0.706) | 41.08 | 0.003 |
| GA + Pruning | 73.96 | 28.25 (55.48) | 77.42 (21.29) | 0.717 (0.555) | **19.28** | 0.014 |
| FT + Pruning | **93.39** | **90.57** (6.84) | **100** (1.29) | 0.094 (0.068) | 458.9 | 0.598 |
| ConMU | $92.15_{\pm 0.52}$ | 88.44 (4.71) | 94.34 (4.37) | $\mathbf{0.116}_{\pm 0.003}$ (0.046) | 51.37 | **0.681** |

# Appendix C   ADDITIONAL EXPERIMENTS

The additional experiments are reported in Table 4, which has the same setup as Table 1.

Received 20 February 2007; revised 12 March 2009; accepted 5 June 2009

