# OpenReview forum: "Breaking the Trilemma of Privacy, Utility, Efficiency via Controllable Machine Unlearning"
_ACM.org/TheWebConf/2024/Conference — TheWebConf24_

### Official Review · Reviewer_dDnn · 2023-11-22

**Novelty:** 5
**Technical Quality:** 5

**Review:**

**Summary:** This paper proposes a method for machine unlearning, where the goal is to modify an existing model to be close to a model trained without a specified subset of the data. The proposed unlearning algorithm in this paper combines three "modules" that can each be tuned:
1. A data selection module, where selecting fewer data points means faster training.
2. Gaussian noise injection, where adding more noise to the forgotten data points means stronger privacy properties.
3. Use of an "unlearning proxy," where the unlearning proxy is a model trained for few epochs on data without the proposed forgotten data.

They empirically evaluate the utility, efficiency, and privacy of unlearning algorithms with their proposed changes.

**Strengths:**
- The proposed algorithm is explained clearly, and the motivation for each proposed module is easy to understand. The authors did a good job explaining the potential outcomes of tuning each of the individual modules (Figure 2 summarizes this well).
- The empirical evaluation seems thorough and includes comparisons of trade-offs in utility, privacy, and efficiency of unlearning algorithms. The comparison of multiple algorithms along all three of these axes seems like a useful contribution.

**Weaknesses:**
- I think the paper could use more explanation of the privacy implications of the progressive Gaussian mechanism. It's not clear to me what exact formulation of privacy the authors are referring to in Section 4.2 when they say, "with more noise being added to the data samples, we will get higher privacy, but lower model accuracy." In experiments, the authors use an "FRM privacy score" metric -- if this is the notion of privacy that the authors refer to with the Gaussian mechanism, then can the authors explain more how the progressive Gaussian mechanism should affect this privacy score in theory?
- As I understand, when the authors say "privacy," they do not mean differential privacy, but instead their own proposed FRM privacy score. Is there any relationship between their proposed measures of privacy and differential privacy? Or, perhaps the authors can give a bit more background on how the word "privacy" is interpreted in machine unlearning.

**Questions:**

See above.

**Reviewer Confidence:**

2: The reviewer is willing to defend the evaluation, but it is likely that the reviewer did not understand parts of the paper

**Scope:**

3: The work is somewhat relevant to the Web and to the track, and is of narrow interest to a sub-community

---

### Official Review · Reviewer_DGEA · 2023-11-23

**Novelty:** 5
**Technical Quality:** 6

**Review:**

**Summary**: The paper studies *”machine unlearning”*—i.e. how to remove the influence of a subset of data points from a given trained model. Given that unlearning faces fundamental tradeoffs between privacy (ability to forget points), utility (accuracy), and efficiency (runtime), the paper designs a framework called *Controllable Machine Unlearning (ConMu)* that offers tunable parameters to control the realized tradeoff between privacy, accuracy, and efficient. The main contribution of this paper is the design and evaluation of this framework.

The framework ConMu combines  several modules with tunable parameters to control the aforementioned tradeoff. The *data selection* module filters which filters points on the basis of mean and standard deviation of the loss (the # of samples can be tuned) with the goal of reducing computation overhead. The *progressive Gaussian mechanism* erases information by continuing to train the original model on noisy samples (the noise can be tuned) with the goal of ensuring privacy. The *fine-tuning with unlearning proxy* partially retrains a new model on the retained dataset (the number of epochs is tunable), and the KL divergence with the unlearned model is computed with the eventual goal of optimizing utility. ConMu combines these modules and optimizes the sum of the cross-entropy loss on a dataset with filtered samples from the retained dataset and filtered noisy samples from the dataset to be forgotten and the KL divergence from the partially retrained model.

The paper empirically evaluates ConMu along the metrics of privacy (using a new proposed score called FRM), utility (test accuracy), and runtime efficiency, in comparison to the baseline approaches of fine-tuning, gradient ascent, and influence unlearning (Table 1 on p.6). The paper focuses on an image classification task for CIFAR-10 on ResNet-18 and CIFAR-100 and SVHN using vgg-16. The experiments illustrate that ConMu achieves a superior tradeoff between privacy, accuracy, and utility than baselines in many settings.

**Strengths:**
- Machine unlearning is an important problem in privacy in machine learning, and the requirements for privacy, accuracy, and efficient differ across different domains. The paper provides an elegant practical framework to all for tunable tradeoffs.
- The empirical analysis of the proposed approach for image classification is quite extensive. The paper considers several important baselines and provides a fairly extensive analysis along privacy, utility, and accuracy.
- The realized tradeoffs between privacy, utility, and accuracy of ConMu are clearly depicted in Figure 3, the comparisons with baselines are clearly demonstrated in Table 1.
- The paper provides a nice qualitative explanation of the importance of each module in Section 5.
- The paper is very well-written and easy to follow. The paper also provides a nice comparison with related work.

**Weaknesses:**
- The empirical analysis is restricted to image classification. It is not clear whether the proposed framework would generalize to other domains with differently structured data and that may be of greater relevance to the WebConf community. That being said, the evaluation on image classification is extensive.
- The paper does not provide an analysis of the properties of the new privacy score (FRM) defined in the paper. While the paper explains that FRM combines 3 existing privacy metrics (Retain accuracy, forget accuracy, and membership inference attacks) and qualitatively explains, it is not clear whether this particular combination of the metrics is the appropriate measure of the privacy. That being said, this is a relatively minor weakness, because the paper paper does provide an analysis of the individual privacy metrics in Table 1.

**Minor comments:**
- Equation (9) parenthesis should be larger

**Questions:**

To what extent would the framework and empirical results  generalize to other domains beyond image classification that may be of greater relevance to the WebConf community?

**Reviewer Confidence:**

3: The reviewer is confident but not certain that the evaluation is correct

**Scope:**

3: The work is somewhat relevant to the Web and to the track, and is of narrow interest to a sub-community

---

### Official Review · Reviewer_r9hp · 2023-11-30

**Novelty:** 6
**Technical Quality:** 6

**Review:**

The authors examine the machine unlearning problem and point out the tension between model utility, forgetting/privacy, and computational cost/efficiency. They propose a new approach involving three components: important data selection, adding Gaussian noise, and fine-tuning an unlearning proxy that allows them to better control the tradeoffs among the competing aims. Empirical results show that such control is achieved and that they're still able to "hit the corners", i.e. are not Pareto-dominated by combinations of other approaches.

I like this work and it is needed in practice. The approaches are not complicated (which is a good thing in my opinion), but have not been put together in this way before. The math looks correct and the empirical results are compelling.

I would appreciate some more discussion about the intuition that adding Gaussian noise leads to catastrophic forgetting and the authors can provide more citations to the same.

Please do one more pass of the manuscript to improve the vocabulary and catch typos.

Not sure what can be done about it, but retrain and retain are so similar words that sometimes I was reading one as the other. Is there some other word than retain that can be used?

In future work, not this paper, it would be interesting to attempt a theoretical analysis and obtain bounds/theoretical characterizations of tradeoffs.

**Questions:**

-

**Ethics Review Description:**

-

**Reviewer Confidence:**

3: The reviewer is confident but not certain that the evaluation is correct

**Scope:**

4: The work is relevant to the Web and to the track, and is of broad interest to the community

---

### Official Review · Reviewer_gYjf · 2023-12-01

**Novelty:** 5
**Technical Quality:** 4

**Review:**

This paper focuses on machine unlearning – the process of updating a machine learning algorithm so that it does not reflect an individual (or set of) data point(s) in order to protect the privacy of individual members of a dataset. In machine learning settings, there is a “trilemma” representing the trade-off between privacy, utility (i.e. accuracy), and efficiency (i.e. runtime). To navigate these trade-offs, the authors put forward “ConMU”, a framework to calibrate MU and navigate MU in a way that coheres with the three conflicting desiderata. Experiments on benchmark datasets suggest that the mechanism proposed outperforms existing methods.

Strengths:
1. The so-called “trilemma” is well motivated and has clear real-world importance
2. Authors provide a framework to systematically control these features
3. Authors provide evidence that their framework and method works on benchmark datasets.

Weaknesses:
1. Under-theorized: Why is the use of ConMU and the particular modules provably superior to FT and the other methods? Or is it somehow reducible to these methods as a particular instance of ConMU?  It would be great to demonstrate analytically that there is some “pareto frontier” along the three dimensions, and your proposed methodology expands this frontier reliably. Instead, some of the empirical results show ConMU being out-performed by FT along certain performance metrics and the description of why remains unclear to me as a reader.
2. It appears that there are numerous cases where FT or another method out-performs ConMU on at least one performance metric. This makes it very difficult to actually evaluate, as a reader, whether ConMU is doing better (it depends on contextual factors determining what desiderata we should care about, as you mention in the intro). The paper would be strengthened if there were very clear-cut, “northstar” performance metrics that show ConMU is pushing the needle and not just one option that trades off certain desirables for others.
3. The best evidence for this (^) seemingly appears in Figure 4. However, it is unclear whether this performance generalizes across a variety of benchmarks, and why the particular benchmark dataset (CIFAR-10 with ResNet-18 under random forgetting) was chosen. Is there some way to capture “how much better” the ConMU is doing instead of just specifying that points “should be” closest to one corner?

**Questions:**

Is the only use case where MU is useful the case of right to be forgotten policy regimes? It might be nice to motivate MU’s other uses, especially if you bring up autonomous driving where RTBF is potentially (?) irrelevant.

Why do you do the results in Figure 4 on only the CIFAR-10 with ResNet-18 under random forgetting scenario? Shouldn’t you do the same test on every scenario represented in Table 1? Also, is there some general way to test that one performed better or to summarize the evaluation across a number of tasks? Or should I just take this one benchmark dataset result as evidence.

In the appendix, in Resnet-18 Class-Wise forgetting (CIFAR-100) MIA/RTE, there’s two underlined in a column and two bolded in another column. Is there a mistake here or am I reading the table wrong?

Theoretically, why should the proposed modules lead to better performance along accuracy, efficiency, and privacy? How could you prove it, or at least motivate it intuitively?

**Reviewer Confidence:**

2: The reviewer is willing to defend the evaluation, but it is likely that the reviewer did not understand parts of the paper

**Scope:**

3: The work is somewhat relevant to the Web and to the track, and is of narrow interest to a sub-community

---

### Official Review · Reviewer_oVP9 · 2023-12-01

**Novelty:** 5
**Technical Quality:** 6

**Review:**

The research introduces Controllable Machine Unlearning (ConMU), a novel framework balancing privacy, model utility, and runtime efficiency, crucial in the field of machine unlearning. ConMU distinguishes itself by integrating three sophisticated modules, demonstrating not only adaptability across different model architectures but also significantly enhancing privacy and efficiency, as evidenced by rigorous testing on benchmark datasets.

Strengths:
- The paper deals with a very timely issue, especially with the responsible web venue. The presentation style including text, tables, and figures is concise and easy to understand.
- ConMU is comprised of three distinct modules: strategic data selection, a progressive Gaussian mechanism, and an unlearning proxy. Together, these elements effectively tackle the complex interplay between privacy, model effectiveness, and processing speed, making this novel framework versatile for a variety of neural-net model structures.
- The efficacy of the ConMU framework was thoroughly tested using comprehensive experiments on benchmark datasets like CIFAR-10 and CIFAR100. These tests highlighted its exceptional ability to maintain privacy while offering a significant boost in processing efficiency compared to standard models.

Weaknesses:
- This work seems more inclined to the core AI framework corresponding to a specific data modality in theory. I think this paper needs to use more practical data and experiments from various modalities and in real-world settings to more fit the Responsible Web track.
- Connected to the first point, while ConMU has shown impressive results in structured test environments using specific image datasets, there are uncertainties regarding how well it would perform in more varied and real-world conditions including various modalities like text and tabular data, bringing its wide-scale applicability into question.

**Questions:**

Please see my comments above, especially with the weak points. Moreover, one critical question might be as follows:

Can we say that adding Gaussian noise guarantees the enhancement of privacy? Maybe adding noises in images and texts affects different in terms of privacy - I think a more thorough and direct investigation is needed to verify this claim.

**Reviewer Confidence:**

3: The reviewer is confident but not certain that the evaluation is correct

**Scope:**

3: The work is somewhat relevant to the Web and to the track, and is of narrow interest to a sub-community

---

### Decision · Program_Chairs · 2024-01-22

**Decision:**

Accept

**Comment:**

Our decision is to accept. Please see the AC's review below and improve the work considering that and the reviewers' feedback for cemera-ready submission. We recommend that the authors expand on the connections to The Web Conference (possibly expanding on the discussions of the relation to the right to be forgotten and GDPR).

"This paper studies machine unlearning - specifically, the goal of removing data points from the training data for privacy purposes. One central challenge in this setting is balancing three objectives (the "trilemma" in this paper): utility (accuracy in general), privacy (having forgotten/removed the effect of the data points we wish to remove) and efficiency (not having to retrain the entire model). This paper proposes an empirical method to address this challenge, with three parts: 1) where data is selected for training 2) Gaussian noise is added to the "forgotten" data points, and 3) an unlearning proxy.

 The reviewers raised questions, such as wanting experiments on a broader range of data types (the authors provided experiments on tabular data during the rebuttal, as well as reproduced variants of Figure 4 for other datasets). The authors also noted that the proposed experiment performs better than other methods on some benchmarks but not all (e.g. Table 1). The authors noted during rebuttal that it may be difficult or impossible for any method to simultaneously perform the best on all approaches. My characterization of the results from Table 1 might be something like "their method often performs comparably to FT + pruning (which is frequently the best) on accuracy and privacy metrics, but substantially outperforms FT + pruning on runtime" - this tradeoff may be a good one for a wide range of settings. Finally, the reviewers also raised questions about theoretical justifications for the paper (e.g. adding Gaussian noise, or the structure more generally). The authors acknowledged that their paper is primarily experimental but said that theoretical justification could be useful for future settings."